# *Corynebacterium striatum*—Got Worse by a Pandemic?

**DOI:** 10.3390/pathogens11060685

**Published:** 2022-06-14

**Authors:** László Orosz, József Sóki, Dávid Kókai, Katalin Burián

**Affiliations:** Albert Szent-Györgyi Health Center, Department of Medical Microbiology, Faculty of Medicine, University of Szeged, H-6725 Szeged, Hungary; soki.jozsef@med.u-szeged.hu (J.S.); kokai.david@med.u-szeged.hu (D.K.); burian.katalin@med.u-szeged.hu (K.B.)

**Keywords:** *Corynebacterium striatum*, antimicrobial resistance, COVID-19 pandemic

## Abstract

The role of *Corynebacterium striatum* has been demonstrated in different nosocomial infections. An increasing number of publications have demonstrated its virulence in the respiratory tract, especially in the immunosuppressed patient population. The number of these patients has increased significantly during the COVID-19 pandemic. For this reason, we aimed to investigate the prevalence and antimicrobial resistance pattern of this species between 2012 and 2021 at the Clinical Center of the University of Szeged, Hungary. Altogether, 498 positive samples were included from 312 patients during the study period. On the isolates, 4529 antibiotic susceptibility tests were performed. Our data revealed that the prevalence of *C. striatum* increased during the COVID-19 pandemic, the rise occurred in respiratory, blood culture, and superficial samples. During the study period, the rifampicin resistance significantly increased, but others have also changed dynamically, including linezolid. The species occurred with diverse and changing co-pathogens in the COVID-19 era. However, the increasing rifampicin and linezolid resistance of *C. striatum* was probably not due to the most commonly isolated co-pathogens. Based on resistance predictions, vancomycin is likely to remain the only effective agent currently in use by 2030.

## 1. Introduction

Corynebacteria are non-spore-forming bacteria with considerable pleomorphism, from club-shaped to long slender bacilli. These bacteria are ubiquitous, especially in soil and water, and some of them are part of the commensal flora of human skin and mucous membranes. To date, more than a hundred species have been identified, and over fifty are associated with human infections [1,2,3].

Of these, the role of *Corynebacterium striatum* (*C. striatum*) is special and controversial. Since this species was first described as a potentially pathogenic bacterium in the 1980s, now is increasingly regarded as an opportunistic pathogen, especially but not exclusively in immunosuppressed patients [4,5,6]. The role of *C. striatum* has been demonstrated in several nosocomial infections. The majority of these first have been limited to respiratory tract infections, isolated bacteremia, central line infections, and occasionally, endocarditis [7,8,9,10,11,12]. However, the scope has broadened recently, to include many other different forms of infection [13,14,15,16,17].

Another concern with the *Corynebacterium* genus is the multidrug resistance of the species. The dissemination of multidrug-resistant *Corynebacterium* species in the hospital environment affects the acquired resistance to β-lactam antimicrobials, clindamycin, erythromycin, ciprofloxacin, and gentamicin [18]. Currently, glycopeptides, linezolid, quinupristin/dalfopristin, daptomycin, and tigecycline are the effective drugs available against *C. striatum* clinical isolates [19,20,21]. Unfortunately, the literature is scarce and related to specific countries. However, the increasing resistance is expected to be a key problem for the genus in the future [21].

In addition, *C. striatum* strains can form biofilms, which contributes significantly to their pathogenicity. Biofilm is a structure that facilitates several bacterial processes affecting virulence and resistance. These include adhesion capacity, metabolite exchange, cellular communication, protection from antimicrobials, and evading host immunity. Thus, biofilm formation facilitates colonization and infections. The *C. striatum* strains get more aggressive in the presence of invasive medical devices, including catheters and endotracheal tubes. Consequently, the formation of bacterial biofilms leads to an increase in healthcare costs, extended hospitalization, and presumably the spread of antibiotic resistance genes [22,23,24,25]. 

Despite these, the respiratory tract infections caused by *C. striatum* have long been an area of controversy. Since the species is a member of the normal human skin and mucous membrane flora, it is often classified as a contaminant in the respiratory tract. However, recently, an increasing number of publications have demonstrated its pathogenicity in the respiratory tract, especially in the immunosuppressed patient population [4,6,15]. The number of these patients has dramatically increased during the years of the COVID-19 pandemic [26,27,28]. In spite of this, the role of *C. striatum* in the light of the pandemic has only been investigated in one publication, and that also only focuses partially on this species using clinical metagenomics methods [4]. For this reason, we aimed to investigate the prevalence and resistance pattern of this species in a retrospective study between 2012 and 2021 at the Clinical Center of the University of Szeged, Hungary. We examined the specimen types in terms of frequency, proportion, and distribution among our departments. Resistance properties were also investigated as part of the study. Moreover, we have also attempted to predict expected resistance trends based on the available data by using a machine learning-based workflow.

## 2. Results

### 2.1. The Number of C. striatum Isolates Has Increased Remarkably over the Last 10 Years

Data on *C. striatum* strains have been available in our *laboratory information* management *system* since 2012. Altogether, 498 positive samples were included from 312 patients between 2012 and 2021. On these isolates, 4529 antibiotic susceptibility tests were performed during the study period, all of which were involved in the analysis.

Plotting these occurrence data according to the year of isolation, a remarkable shift can be seen (Figure 1A). The number of isolates has increased significantly since 2018. In 2019, double the amount isolated in 2018 was detected, followed by 38 and 37 more per year until 2021 (Figure 1A). At the same time, it is also noticeable that the proportion of positive samples has also increased remarkably since 2019 (Figure 1B). To make it more comparable, the period between 2012 and 2021 was divided into two parts: the pre-COVID-19 (2012–2019) and the COVID-19 eras (2020–2021). In this split, this comparison resulted in a significant difference between the pre-COVID-19 and COVID-19 years (Figure 1C).

These data suggest that the number of *C. striatum* strains has increased remarkably over the last 10 years, especially since 2019 and during the COVID-19 era years (2020–2021).

### 2.2. C. striatum Was Significantly more Frequent in Some Sample Types in the COVID-19 Era

To gain a more in-depth understanding of the sources of the *C. striatum* strains, the isolates were also analyzed by sample type. The main clinical specimen types were identified as respiratory, blood culture, deep tissue, superficial, and urine samples. During the comparison of the positivity of each sample type, it was noticeable that the number of *C. striatum* isolates increased within the total number of positive specimens (Figure 2A).

To present the trends more clearly, the proportion of *C. striatum*-positive samples within each sample type was plotted separately (Figure 2B). This plot shows a steady increase in the rate of respiratory, superficial, and deep isolates, especially since 2019. A much milder upward trend can also be seen for blood cultures.

In the next phase of our work, we investigated whether the COVID-19 pandemic caused a significant change in the number of *C. striatum* isolates. To put this into perspective, we have also used here the division of the study period into pre-COVID-19 and COVID-19 years. Significant increases can be seen for respiratory, blood culture, and superficial samples (Figure 2C).

These data together suggest that *C. striatum* was significantly more frequently isolated from some sample types in the COVID-19 era.

### 2.3. C. striatum Strains Were Consistently Present in Some Wards, including COVID-19 Care Sites

We also considered it important to examine from which departments and in what numbers *C. striatum* strains were isolated between 2012 and 2021 at the Clinical Center of the University of Szeged. Our analysis showed that the species was present continuously in some clinics (e.g., otorhinolaryngology, surgery, and oncology) for almost the entire study period (Figure 3, black arrows). At the same time, it is important to highlight that in other departments receiving mainly COVID-19 patients (e.g., infectology and special COVID-19 care sites), the number of isolates has only increased substantially in the last two years (Figure 3, red arrows). An interesting exception was the Department of the 1st Internal Medicine, where the species has been present continuously since 2018, but in 2021 the number of isolates increased significantly (Figure 3, gray arrow). This may be explained by the fact that this building has also been used for COVID-19 care since the end of 2020.

These results support the concept that COVID-19 may have contributed considerably to the increase of *C. striatum* isolates. Unfortunately, as the strains were not preserved, it was not possible to examine for potential clonality by genetic methods.

### 2.4. One-Third of C. striatum Strains Were Isolated from Cases Associated with Viral Infection in 2020 and 2021

To support our hypothesis that *C. striatum* was isolated more frequently in patients suffering from COVID-19, we also examined the referral diagnosis of the positive samples received in the microbiology laboratory. A total of 170 isolates, for which sufficient such information was available, were involved in this analysis from 2020 and 2021. We have associated these with the referral diagnoses according to their frequency (Table 1). The first two most common referral diagnoses were related to viral infection. Moreover, other such diagnoses were also found (Table 1). To make it even more explanatory, the submitting diagnoses were divided into two groups according to whether they were presumably related to COVID-19. Table 1 shows that 30% of the isolates were from such cases.

Based on these results, it can be stated that one-third of the *C. striatum* isolates were derived from suspected COVID-19 cases in 2020 and 2021.

### 2.5. There Was No Significant Change in the Antibiotic Susceptibility of C. striatum Isolates between 2012 and 2021

We then examined how the antibiotic susceptibility of the isolates changed over the years. In line with the guidelines of the European *Committee* on Antimicrobial Susceptibility Testing (EUCAST), the range of antibiotics tested included ciprofloxacin, moxifloxacin, clindamycin, gentamicin, linezolid, rifampicin, tetracycline, and vancomycin. We have aimed to calculate a cumulative antimicrobial resistance index (ARI) as a possible tool to monitor the antimicrobial resistance trend. To calculate the ARI, the model for measuring antibiotic resistance used by De Socio et al. was followed [29].

The values obtained were plotted chronologically on a line graph (Figure 4A). The data demonstrate that the ARI values of some drugs varied dynamically (e.g., rifampicin), while others remained consistently high (e.g., fluoroquinolones) or low (e.g., vancomycin) between 2012 and 2021.

The ARI values of each antibiotic were also analyzed individually, and the mean values with their standard deviations of the pre-COVID-19 and COVID-19 eras were compared (Figure 4B). No significant change was detected for most agents, except for rifampicin, where a significant increase in resistance was observed in the years of the COVID-19 era (*p* < 0.05).

We then compared the average ARI values of all examined antibiotics in the pre-COVID-19 and COVID-19 eras (Figure 4C). In line with the previous observations seen in Figure 4B, this comparison showed that there was no significant change in ARI values despite the increased number of *C. striatum* isolates associated with the COVID-19 pandemic.

To further understand the potential changes for each antibiotic, the slopes of the ARI curves for each agent were also calculated and plotted (Figure 5A). This representation can show in more detail the changes in resistance in the pre-COVID-19 and COVID-19 periods, where positive slope values indicate the growth of resistance and negative values indicate progress towards susceptibility. As can be seen in Figure 5A, there was indeed no remarkable change for most agents. The differences for ciprofloxacin, moxifloxacin, clindamycin and vancomycin were small and in the same direction over the two periods. Slightly larger differences were observed for gentamicin and tetracycline. However, tetracycline appears to change in the same direction, as opposed to gentamicin, which changes in opposite directions depending on the pre-COVID-19 and COVID-19 periods. The largest differences were detected for linezolid and rifampicin. The former showed a remarkable and opposite trend in the two periods, while the latter also showed a notable but parallel tendency. Using the same method, we also examined the resulting slope for each agent, by summing the values measured in each period (Figure 5B). From this plot, it is clear that the most noteworthy difference in the direction of resistance is seen for rifampicin. This is followed in magnitude by the change in tetracycline, but with the opposite sign. The next in this direction is gentamicin, and then moxifloxacin with a slight change in the direction of resistance (Figure 5B). These results are in good agreement with the resistance trends seen in Figure 4A,B, but better express their relative magnitude.

These data together suggest that, although there may have been dynamic changes for individual drugs over the period, there were no significant alterations in antibiotic resistance, except for rifampicin. The slight upward trend in resistance observed with rifampicin draws attention to the fact that the empirical use of this drug requires increased caution on the part of our clinicians.

### 2.6. Predicted Theoretical ARI Values for Currently Used Agents Project Two Trajectories until 2030

We attempted to outline trends in antimicrobial resistance for currently used agents over the next 10 years using the Pycaret machine learning-based software package. The theoretical ARI values generated this way can exceed 1. These values were calculated by using only the currently existing influencing factors, and do not take into consideration unexpected events that may occur in the future (e.g., the development of a new effective antibiotic or the outbreak of a new pandemic).

The resulting ARI values predict two groups of trajectories, which can be distinguished by their slope (Figure 6). Trajectory #1 indicates increasing resistance to linezolid with a slope of 0.13 and rifampicin with a slope of 0.03. Trajectory #2 groups those agents (ciprofloxacin, moxifloxacin clindamycin, gentamicin, and tetracycline) that have no alterations in their resistance (slope: 0.00). Due to the low ARI values of vancomycin resistance, the Pycaret software could not provide a prediction for this drug. However, based on the available data, the resistance to vancomycin is not expected to change significantly in the foreseeable future.

Based on these results, major changes in resistance are expected for linezolid and rifampicin over the next 10 years.

### 2.7. The C. striatum Occurred with Diverse and Changing Trends of Co-Pathogens in the COVID-19 Era

In the next phase of our work, we examined the pathogens with which *C. striatum* was isolated simultaneously from different samples. Again, the time of isolation was divided into pre-COVID-19 and COVID-19 periods (Figure 7A). First, we examined the relationship between the number of *C. striatum* strains and the total number of co-pathogenic bacteria over the two stages. It was found that both the number of *C. striatum* isolates and the prevalence of co-pathogenic bacteria were significantly higher in the COVID-19 era (Figure 7A). To better compare the prevalence of the two groups, we determined the pre-COVID/COVID-19 prevalence ratios for both (Figure 7B). This suggests that the occurrence of *C. striatum* with co-pathogens was 41% more frequent than *C. striatum* alone.

Subsequently, we also examined in detail the prevalence of each co-pathogen (Figure 8A). The most frequently isolated co-pathogen in the COVID-19 era was *Staphylococcus aureus*, which represents a significant change compared to the pre-COVID-19 period. Similar noteworthy changes were observed in the prevalence of *Enterococcus faecalis*, *Enterococcus faecium*, and *Candida albicans*. There was also a smaller but still significant increase in the occurrence of *Acinetobacter baumannii*, *Candida parapsilosis*, *Morganella morganii*, and *Raoultella ornithinolytica* concomitantly with *C. striatum* in samples taken during the COVID-19 period (Figure 8A).

Then, we also examined the samples from which co-pathogens were most frequently isolated concomitantly with *C. striatum* (Figure 8B). The co-pathogens were most commonly isolated from tracheal, blood culture, and wound specimens in 2020 and 2021. In these sample types, the prevalence of co-pathogens was also significantly increased (Figure 8B). A noteworthy but much smaller difference is also seen for the catheter-derived samples. Figure 8B also shows that the overall number of positive samples increased significantly during the COVID-19 era.

Taken together, these data thus show a significant change in the prevalence of *C. striatum* co-pathogens during the COVID-19 era.

### 2.8. The Rifampicin and Linezolid Resistance of C. striatum Is Probably Not Due to the Most Commonly Isolated Co-Pathogenic Bacteria

To shed light on whether the source of the continuously increasing rifampicin resistance in *C. striatum* might be a co-pathogen, we investigated the progression of rifampicin resistance in the most commonly co-isolated species. Of the most frequently isolated co-pathogens, only *Staphylococcus aureus* is required by EUCAST to be tested for rifampicin. According to our results, although the rifampicin resistance of *C. striatum* increased steeply over the years with several breaks, we did not observe this phenomenon in the case of *S. aureus* (Figure 9A). The value of Spearman’s rank correlation coefficient calculated from the data is also close to zero, which indicates that there is no correlation between the two phenomena.

For linezolid, EUCAST requires testing not only for *S. aureus* but also for Enterococci. Correlations to this agent were therefore tested for these bacteria (Figure 9B). In this case, we also calculated Spearman’s rank correlation coefficient for each species. The results obtained were close to zero in all cases.

This suggests that the source of rifampicin and linezolid resistance in *C. striatum* is probably not the most frequently isolated co-pathogenic bacteria.

## 3. Discussion

*C. striatum* is accepted nowadays as an opportunistic pathogen, but its role is still controversial since it is a member of the normal flora of several parts of the human body [3]. Nevertheless, recently, more and more reports have confirmed its pathogenicity, especially in hospital settings and in immunosuppressed patients [5,6,9,14]. Since the COVID-19 pandemic has significantly increased the number of hospitalized and even ventilated patients and has been shown to cause immune system-disruption effects, it seemed important to investigate the impact of the pandemic on the incidence of this opportunistic bacterium. To the authors’ knowledge, only one publication has examined this issue and detected a hospital outbreak involving 14 patients across three COVID-19 intensive care units [4]. For this reason, we have investigated the prevalence and resistance characteristics of this species at the Clinical Center of the University of Szeged between 2012 and 2021, with a special focus on the impact of the COVID-19 pandemic.

Our results have shown that *C. striatum* isolates have become increasingly common in our routine clinical microbiology practice, especially since 2019 and during the COVID-19 era years (Figure 1). It is important to underline that there have been no changes in the official diagnostic protocol for corynebacteria between 2012 and 2021 in our department. However, it cannot be excluded that systematic study of the literature has led colleagues to increasingly classify this bacterium as a pathogen. This trend is well in line with the fact that a growing number of cases of invasive infections have been reported worldwide [18].

Among the different types of *C. striatum*-positive specimens, the number of respiratory, superficial, and deep samples, and with a much milder upward trend the blood cultures were increasing (Figure 2A,B). This also correlates well with literature data, as this species is commonly seen in infections caused by invasive medical devices [5,18]. In the context of the COVID-19 pandemic, the number of such patients has increased significantly worldwide [30]. This probably creates favorable conditions for the development of nosocomial infections caused by *C. striatum*. In line with this, broken down the study years into pre-COVID-19 and COVID-19 periods, significant increases were seen for respiratory, blood culture, and superficial samples in the COVID-19 era (Figure 2C). The authors could not find such evaluation in the literature.

As hospital outbreaks have been described earlier [4,22,23], we also investigated which departments of the Clinical Center of the University of Szeged had the highest incidence of *C. striatum* between 2012 and 2021. Our analysis demonstrated that this species was present continuously in some clinics for almost the entire study period (Figure 3, black arrows). At the same time, it is important to highlight that in other departments receiving mainly COVID-19 patients, the number of isolates has only increased substantially in the last two years *(*Figure 3, red arrows). This raises the possibility that a strain, which was present before the COVID-19 pandemic, has spread in the Clinical Center. Unfortunately, since the strains were not preserved, we could not examine their clonality by genetic methods.

To somehow support this hypothesis, we also examined the referral diagnoses of the *C. striatum*-positive cases in 2020 and 2021 (Table 1). Our analysis demonstrated that one-third of the strains were isolated from suspected COVID-19 cases (Table 1). The COVID-19-related submitting diagnoses were: viral pneumonia, viral infection, dyspnoea, coronavirus infection, severe respiratory failure, other viral pneumonia, and COVID-19 with the detected virus. These all suggest a need for ventilation in patients, which creates favorable conditions for secondary bacterial infections, including respiratory tract infections caused by *C. striatum*. This shows that in wards caring for COVID-19 patients, optimal conditions for this opportunistic respiratory bacterium were discovered. However, a clear description of this risk is still hard to find in the literature [4,31]. The fundamental goal of this work is to raise awareness of this threat.

With such an emerging pathogen, the question of treatability, i.e., antibiotic resistance, is also important. Non-diphtheriae corynebacteria often present a therapeutic difficulty to the physicians [32]. Infections caused by these opportunistic *Corynebacterium* spp. generally characteristic of patients suffering from immunodeficiency. This includes disorders of bone marrow activity, cancer, post-surgery, invasive diagnostic procedures, and AIDS. The risk of infection is also increased by long-term hospitalization, antibiotic therapy, radiotherapy, and treatment with cytostatics or steroids [33]. We consider it important to highlight that steroid therapy is also widely used in the therapy of COVID-19 [34]. Recent studies characterizing the resistance of corynebacteria draw attention to the most frequently occurring mechanisms [32,33,35,36,37,38,39]. The results show the participation of extrachromosomal genetic elements in the transmission of resistance genes in both pathogenic and opportunistic species [33]. Antibiotic resistance genes are often located on large plasmids and transposons [33].

In the present study, eight agents were tested for susceptibility continuously according to EUCAST standards between 2012 and 2021 [40]. For most of these, dynamically varying resistance landscapes were found (Figure 4A). Over the years, most isolates have shown consistent resistance to fluoroquinolones. This type of resistance is based on *gyrA* mutations in the genus *Corynebacterium* [21,32,33,39]. Similar high levels of resistance were observed with clindamycin [32]. The MLS phenotype (resistance to Macrolides, Lincosamides, and Streptogramin) is a mechanism also frequently observed in different *Corynebacterium* spp., including *C. striatum* [21,39]. It is linked with the occurrence of three different mechanisms: modification of the ribosome binding site associated with methylation or mutation, active efflux, and enzymatic inactivation of the drug. The first two MLS resistance mechanisms are of the highest importance [33]. Of the other antibiotics tested, tetracycline and linezolid have shown remarkable fluctuations in amplitude over the years. Regarding tetracycline, *C. striatum* was found resistant in 97% in an earlier study [41]. These data were confirmed by the detection of the gene *tetM*, responsible for resistance to all tetracyclines, which is due to ribosomal protection [33]. The intermittent emergence of resistance to linezolid is particularly worrying because it is often the first antibiotic of choice in the therapy of multidrug-resistant *C. striatum* strains [42]. Although the genetic background of resistance to this agent is already known in other species, the authors found no data in the literature on linezolid resistance among corynebacteria. For this reason, it is particularly unfortunate that strains previously expressing this phenotype have not been preserved for genetic analysis. In other bacteria, the mutation of 23S rRNA has been established as one of the resistance mechanisms. Furthermore, the transferable modification of 23S rRNA by the *Cfr* methyltransferase can also cause high resistance to linezolid [43]. The only significant change in the years of the COVID-19 pandemic is the considerable increase in rifampicin resistance, which started from almost zero in 2012 and reached 60% by 2021 (Figure 4A,B). However, it is important to note that between 2012 and 2014, fewer than twenty *C. striatum* isolates were tested per year. Despite all this, it is also quite a worrying phenomenon, since on the one hand values of around 25% have been previously reported [38], and on the other hand the drug is often administered in combination specifically for the treatment of *C. striatum* [39,44,45]. This resistance is nearly always due to a genetic change in the β subunit of RNA polymerase (RpoB) [39]. Despite all these negative changes, the cumulative antibiotic resistance index did not increase significantly in the COVID-19 era (Figure 4C). These data suggest that pre-pandemic strains have survived and spread in our Clinical center.

To better understand the changes in resistance, we examined the slope of the ARI curves for each antibiotic in the pre-COVID-19 and COVID-19 eras. In most cases, we found that the ARI slope values of the agents hardly changed between the two periods (Figure 5A). Of the eight antibiotics tested, two showed remarkable changes: linezolid displayed a counterbalancing trend in the opposite direction, while rifampicin, unfortunately, presented a trend towards resistance in both periods (Figure 5A). To make the result of the changes over the two periods even clearer, the fluctuations in the slopes of the ARI curves have been summarized (Figure 5B). This type of visualization further highlights that the most worrying increase in resistance, of almost 15 percent, occurred with rifampicin. To the authors’ knowledge, this is the first study to describe these relationships in such detail.

In order to get a more nuanced picture of the evolution of resistance for each agent, we predicted the slopes of the theoretical ARI curves up to 2030 (Figure 6). By magnifying the differences in this way, the drugs tested were divided into two groups. The first trajectory was the steepest ascent and represents linezolid and rifampicin. This phenomenon is likely to cause problems in the future. According to literature data, rifampicin and linezolid are the agents, along with vancomycin, least likely to develop resistance [39]. For this reason, the tendency of increasing rifampicin resistance is worrying. Although our model also predicted a significant near-term increase in resistance for linezolid, no precedent for this has been found in the literature. It will be an interesting development shortly to see if this assumption proves true.

The path of the second group of agents (ciprofloxacin, moxifloxacin clindamycin, gentamicin, and tetracycline) showed no alterations in their resistance. Unfortunately, for fluoroquinolones, a constant level of resistance has stabilized (Figure 6). This value is slightly lower for tetracycline (ARI around 0.8), but even for gentamicin, it is around 0.5 (Figure 6). Regrettably, gentamicin is no longer recommended for testing in the current version of EUCAST as there is insufficient evidence of its efficacy for corynebacteria. Thus, the most suitable drug for use remains vancomycin, for which the proportion of resistant isolates was so low that the Pycaret software could not even predict its future development (Figure 6). However, the resistance to this drug is not expected to change significantly in the foreseeable future. To the best of the authors’ knowledge, this is the first such analysis of the possible future evolution of *C. striatum* resistance. In light of the dwindling therapeutic choices, new antimicrobials must be tested and may be added to the therapeutic arsenal. An excellent example of this is a recent study by Folliero et al., who proposed anthelmintic drugs to counter *C. striatum* infections [46].

To gain a more comprehensive picture of the situation regarding the increase in the prevalence of *C. striatum* isolates, we examined which other pathogens were most frequently isolated with the species. In this case, we also applied the breakdown into pre-COVID-19 and COVID-19 periods (Figure 7). We have analyzed the occurrence of *C. striatum* alone and with co-pathogenic bacteria (Figure 7A). Both the *C. striatum* and the co-pathogens showed a significant increase in the number of isolates during the years of the COVID-19 era. To compare the extent of the increase, we also calculated and plotted the ratio of the number of strains isolated in both cases in the COVID-19 and pre-COVID-19 era years (Figure 7B). The comparison shows that this ratio is 41% higher for co-pathogens, therefore *C. striatum* was isolated more often with other pathogenic bacteria than alone. To the authors’ knowledge, this is the first study to describe such a comparison.

In the next phase of our work, we also examined in detail, which pathogenic bacteria co-occurred with *C. striatum* in each sample type and how often. The co-occurrence of *S. aureus*, *E. faecalis*, *E. faecium*, and *C. albicans* with *C. striatum* increased significantly in the COVID-19 era (Figure 8A). Concerning sample types, the same comparison showed that co-pathogens were most frequently detected in samples of the respiratory tract, blood culture, and wound origin (Figure 8B). In catheter, surgical, abscess, and pleural punctate samples, co-pathogens were isolated rarely (Figure 8B). This qualitative distribution of specimen type frequencies does not show noteworthy differences from the literature data [18]. Nonetheless, to the best of the authors’ knowledge, no such quantitative analysis has yet been published.

Given the expected rise in resistance of rifampicin and linezolid are of concern, we have also investigated whether this phenomenon could be linked to the common co-pathogens. Although the ARI value for *C. striatum* rifampicin has been fluctuating, this visualization also shows an increasing trend, which rose sharply since 2018 (Figure 9A). However, a similar magnitude of change was not observed for *S. aureus*, and even, the decline seen in *C. striatum* rifampicin resistance was the time of the most significant increase in the case of *S. aureus* (Figure 9A). All these data demonstrate that *S. aureus* remained sensitive to rifampicin over the period studied. The Spearman’s rank correlation coefficient calculated for the two species does not reflect any relationship either. No significant correlation was found between the ARI values of linezolid in the case of *C. striatum* and the corresponding ARI values of *S. aureus*, *E. faecalis*, or *E. faecium* during the period studied (Figure 9B). These data suggest that the sources of resistance for the two agents were not the co-pathogens, but some other source. We hypothesize that the increased resistance to these agents may also be due to their increased use in our healthcare. This is supported by the data that the number of nosocomial infections that may require the administration of linezolid and/or rifampicin is rising in the COVID-19 era [28,47,48].

Together, these data suggest that as a consequence of the COVID-19 pandemic, the incidence of *C. striatum* and its co-pathogens has increased remarkably among patients at the University of Szeged, and the spectrum of agents available for treatment is narrowing. The primary purpose of this work is to raise awareness about this issue.

## 4. Materials and Methods

### 4.1. Study Setting

The present retrospective microbiological study was carried out using data collected, corresponding to the period between 1 January 2012 and 31 December 2021, at the Department of Medical Microbiology, University of Szeged, Hungary. This clinical microbiology laboratory serves the Albert Szent-Györgyi Clinical center, which is an 1800-bed primary- and tertiary-care teaching hospital in the Southern Great Plain of Hungary. The protocol for collecting and evaluating samples did not change during the study.

### 4.2. Microbiological Data Set

This study was conducted using local data that were exported from the clinical microbiology laboratory information system (MedBakter, Asseco Central Europe Ltd., Budapest, Hungary), and was reported into a customized database. Data included the types of specimen, species of isolates, antimicrobial susceptibility patterns, and the referral diagnoses of the patients. Antimicrobial susceptibility testing results were always determined and interpreted according to the current EUCAST breakpoints [40].

### 4.3. Calculating Antibiotic Resistance Index

In all cases, antibiotic susceptibility testing was performed by the disc diffusion method as described in the EUCAST manual [49]. To calculate the ARI, the model for measuring antibiotic resistance in ESKAPE pathogens used by De Socio et al. [8] was followed. Briefly, for each antibiotic tested, a score of 0 for susceptibility, 0.5 for intermediate resistance, or 1 for resistance were assigned, and the ARI was calculated by dividing the sum of these scores by the number of antibiotics tested, giving a maximum score of 1. Thus, an ARI of 0 corresponded to a pan drug-susceptible organism and an ARI of 1 to a pan drug-resistant strain.

### 4.4. Statistical Analysis

The data were exported from the laboratory information system into the MS Excel 2016 (Microsoft Corp., Redmond, WA, USA) and GraphPad Prism version 8 (GraphPad Software, San Diego, CA, USA) software. The MS Excel 2016 was used to store the data, and its SLOPE function was used to determine the slope values of different ARI curves. The Graphpad Prism 8 was used for statistical analysis and plotting. All values are expressed as means ± standard deviation, where appropriate. Welch’s *t*-test or Mann-Whitney U test was used to compare the means of populations. Chi-square test or Fischer’s exact test was used for frequency distributions. *p* values < 0.05 were considered statistically significant. The Spearman nonparametric correlation function of the GraphPad Prism 8 was used to determine the correlation between the ARI values of rifampicin and linezolid between *C. striatum* and the co-pathogens.

### 4.5. Time Series Forecasting Using Machine Learning

Predictions of ARIs were performed in Python 3.10. using Pycaret [50]. Pycaret is a Python wrapper for several machine learning libraries and frameworks including scikit-learn, XGBoost, LightGMB, CatBoost, spaCy, Optuna, Hyperopt, and Ray. The package allows the user to automate machine learning workflows. In the prediction process, the data was split into 70% test set and 30% train set. The forecasting approach was univariate, without using Exogenous Variables. Expanding Windows Splitter was used as cross-validation. The data set frequency was yearly. On each dataset, 24 models were analyzed, then based on symmetric mean absolute percentage error (SMAPE) the best model was chosen, and hyper tuned for 1000 iterations. Thereafter, the model predicted ahead of 9 years. For each prediction, the exact model can be found in Appendix A.

## 5. Conclusions

Overall, this study aiming to shed light on the impact of the COVID-19 pandemic on the prevalence and resistance changes of *C. striatum* isolates revealed that this species was significantly more commonly isolated in 2020–2021 than between 2012 and 2019. This rise occurred in respiratory, blood culture, deep, and superficial samples. Rifampicin resistance was significantly increased, the source of which was probably not the most frequently isolated co-pathogen. The theoretical ARI prediction projected two groups of possible future trajectories, showing the increasing resistance of rifampicin and linezolid, and that vancomycin is likely to remain the only effective agent currently in use. Exploring the causes and consequences of these changes could be the subject of interesting future studies.

## Figures and Tables

**Figure 1 pathogens-11-00685-f001:**
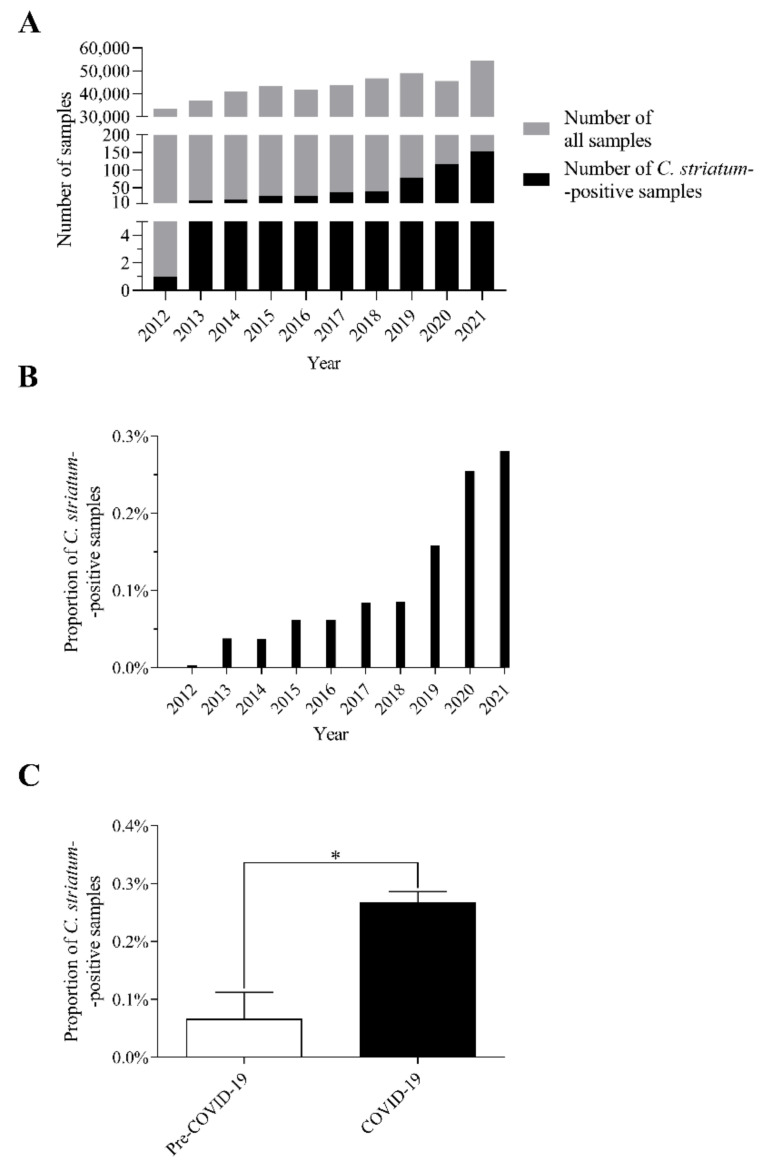
The number and proportion of *C. striatum*-positive samples between 2012 and 2021. (**A**) The number of all positive samples and those from which *C. striatum* was isolated between 2012 and 2021. (**B**) The proportion of *C. striatum*-positive clinical specimens in the same period. (**C**) Comparison of the percentage of *C. striatum*-positive samples in pre-COVID-19 and COVID-19 eras (* *p* < 0.05). The bars represent means and standard deviation indicated at the top of the columns, where appropriate.

**Figure 2 pathogens-11-00685-f002:**
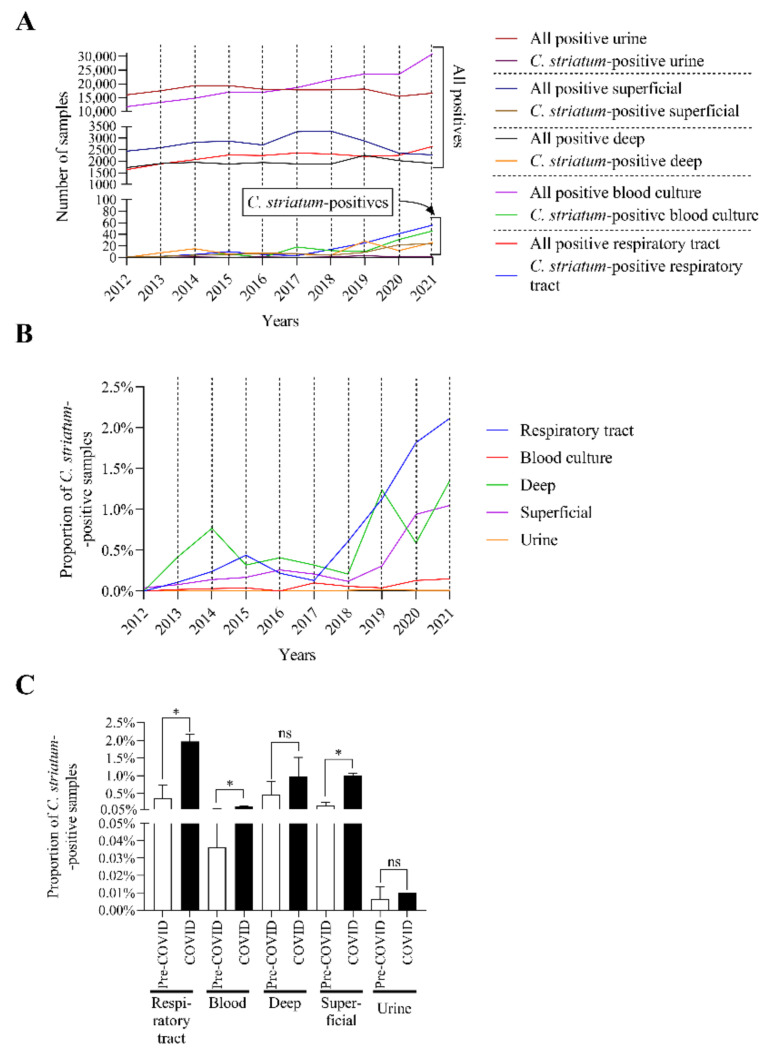
The number, distribution, and proportion of *C. striatum*-positive samples among the different sample types in the pre-COVID-19 and COVID-19 eras. (**A**) The number of all positive samples by type and those from which *C. striatum* was isolated between 2012 and 2021. (**B**) Comparison of the *C. striatum* positivity proportion among the main specimen groups in the same period. (**C**) Comparison of the proportion of *C. striatum*-positive sample types in pre-COVID-19 and COVID-19 eras (* *p* < 0.05). The bars represent the means and standard deviation indicated at the top of the columns.

**Figure 3 pathogens-11-00685-f003:**
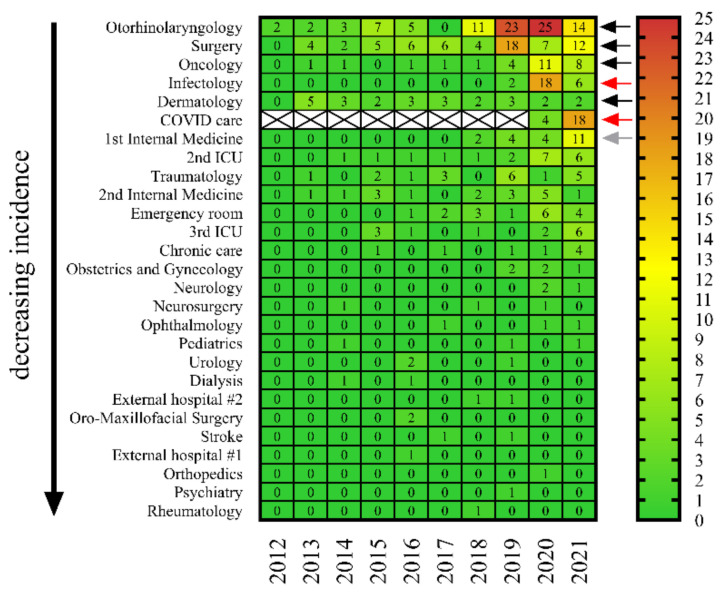
The prevalence of *C. striatum* in the departments of the University of Szeged between 2012 and 2021. The colors of the cells and the numbers in them indicate the number of isolates as seen on the scale. (Black arrows: continuous incidence during the study period. Red arrows: increased incidence in the last two years. Grey arrow: case accumulation observed in the Department of the 1st Internal Medicine [see text for details]). Strikethrough cells indicate that there was no COVID care before 2020.

**Figure 4 pathogens-11-00685-f004:**
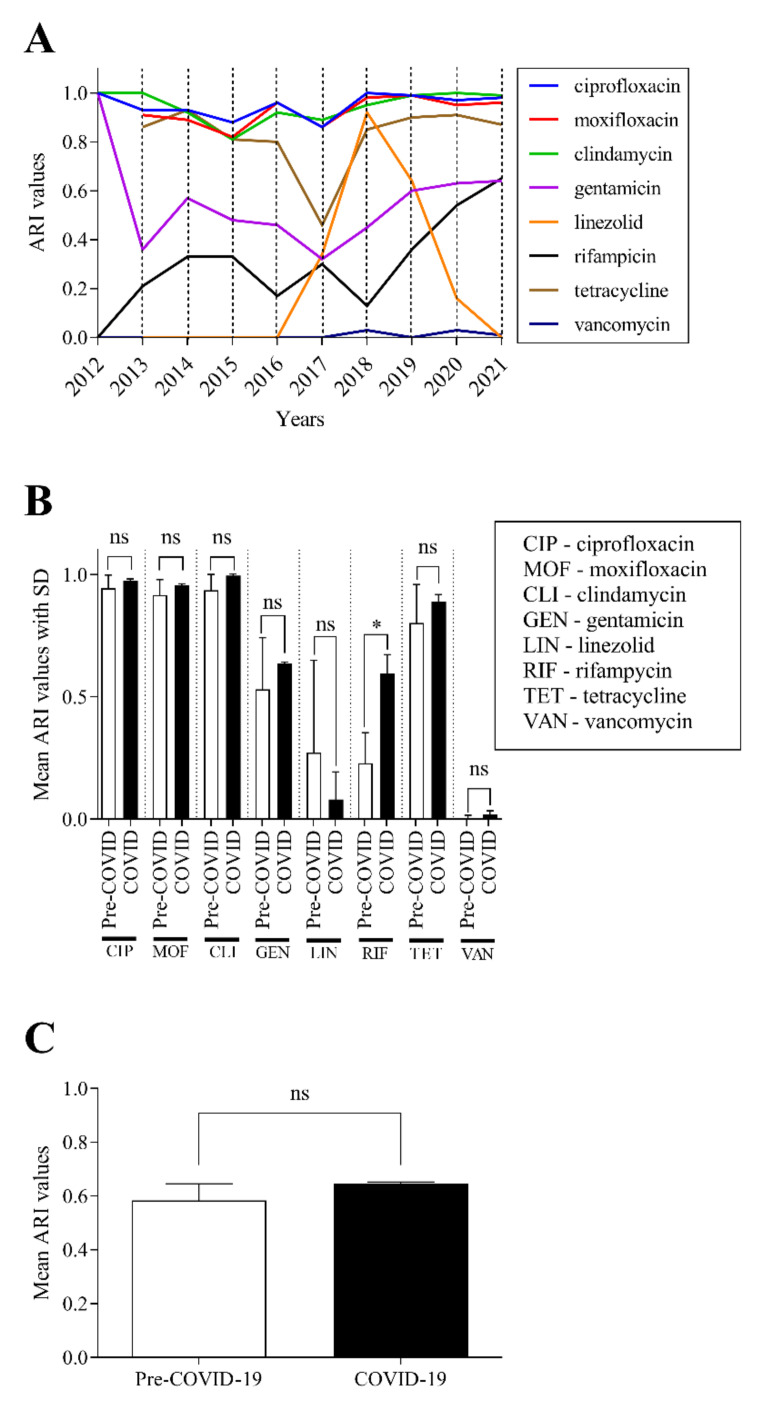
Changes in susceptibility of tested antibiotics between 2012 and 2021. (**A**) Cumulative antibiotic resistance index (ARI) of the antibiotics tested between 2012 and 2021. (**B**) Evaluation of ARI values of each tested antibiotic in the pre-COVID-19 and the COVID-19 eras. (**C**) Comparison of the sum of mean ARI values in the pre-COVID-19 and the COVID-19 periods. * *p* < 0.05. The bars represent the means and the standard deviation indicated at the top of the columns.

**Figure 5 pathogens-11-00685-f005:**
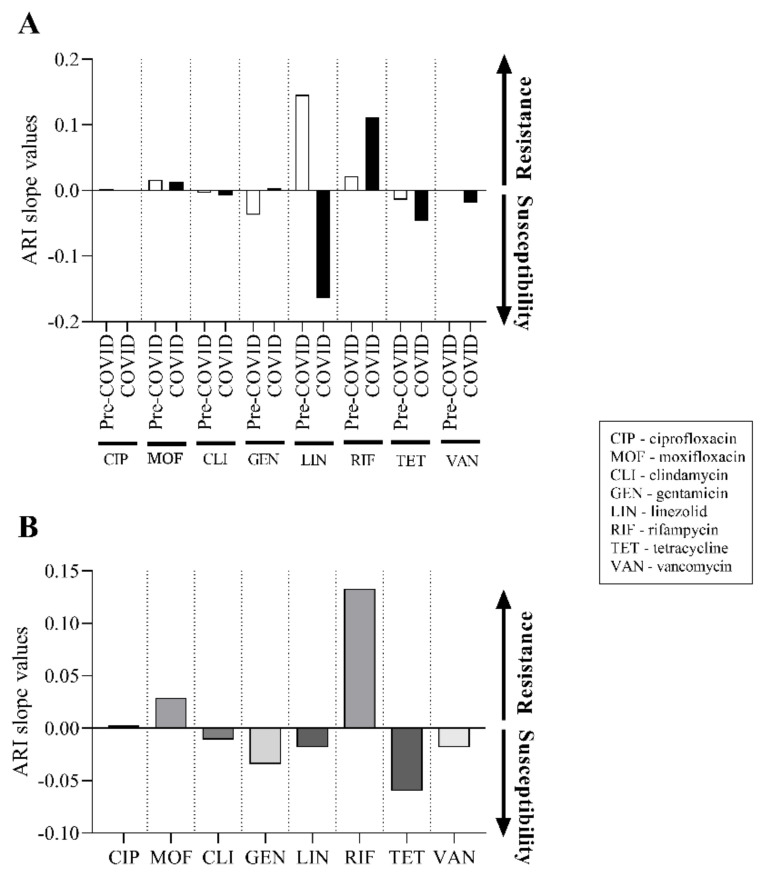
The changes in the slope of ARI curves of tested antibiotics in the pre-COVID-19 and COVID-19 periods. (**A**) ARI curve slope values of each tested drug in the years of the pre-COVID-19 and the COVID-19 eras. (**B**) Resulting slope values of ARI curves of each tested antibiotic in the entire study period. Arrows indicate the orientation of changes in the direction of resistance or susceptibility.

**Figure 6 pathogens-11-00685-f006:**
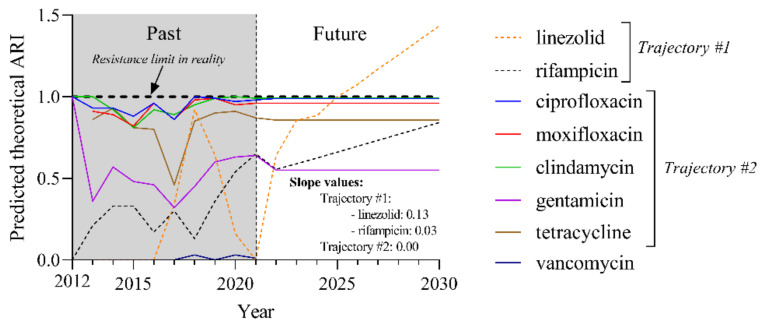
The predicted theoretical ARI values for currently used agents until 2030. Trajectory #1 antibiotics are indicated by dashed lines and trajectory #2 antibiotics by solid lines.

**Figure 7 pathogens-11-00685-f007:**
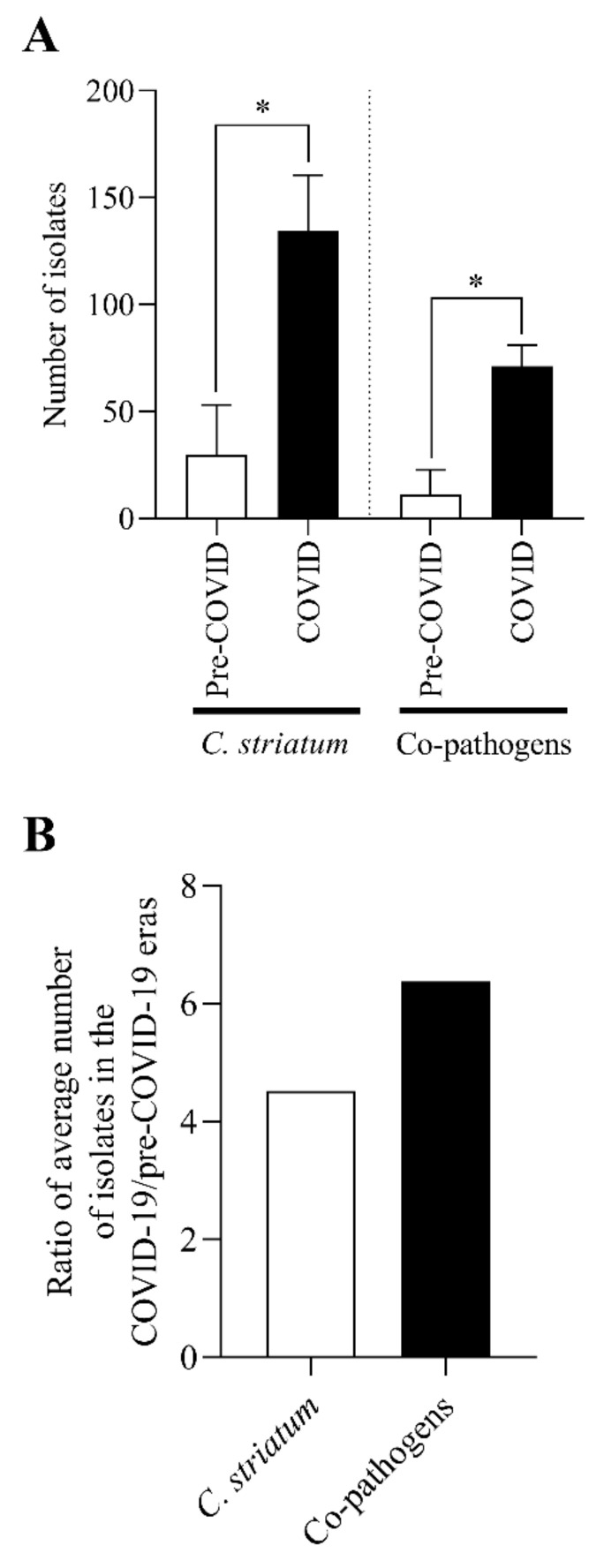
Analysis of the occurrence of *C. striatum* alone and with co-pathogenic bacteria. (**A**) Prevalence of *C. striatum* alone and with co-pathogenic bacteria in pre-COVID-19 and COVID-19 eras. (**B**) The ratio of the mean of *C. striatum* isolates alone or in co-occurrence with other pathogenic bacteria. * *p* < 0.05. The bars represent means and standard deviation indicated at the top of the columns, where appropriate.

**Figure 8 pathogens-11-00685-f008:**
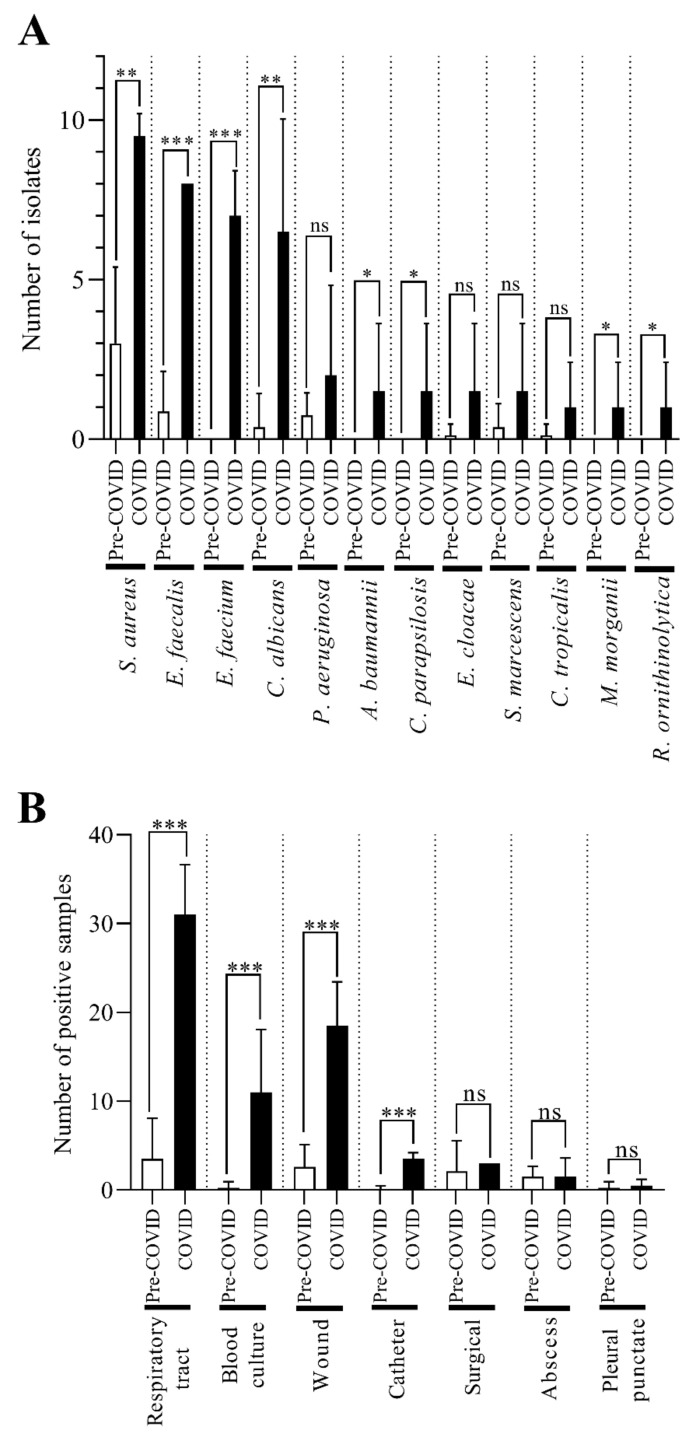
The prevalence of pathogens isolated in association with *C. striatum* in the pre-COVID-19 and COVID-19 eras. (**A**) Prevalence of co-pathogens with *C. striatum* in the pre-COVID-19 and COVID-19 periods. (**B**) Frequency of different sample types in which the co-pathogens occurred at the same times. * *p* < 0.05; ** *p* < 0.01; *** *p* < 0.001. The bars represent the means and standard deviation indicated at the top of the columns.

**Figure 9 pathogens-11-00685-f009:**
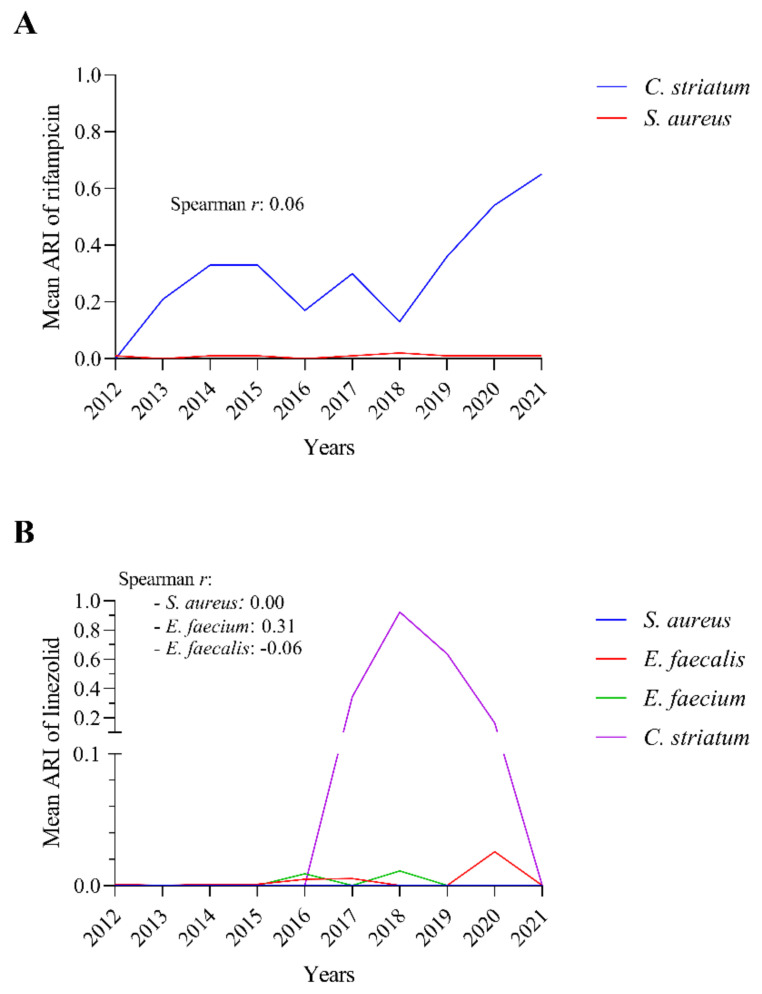
The trends in the mean ARIs of rifampicin and linezolid in the cases of *C. striatum* and co-pathogens between 2012 and 2021 and the calculated Spearman *r* values. (**A**) Mean ARI values of rifampicin for *C. striatum* and *S. aureus* and the Spearman *r* value. (**B**) Mean ARI values of linezolid for *C. striatum*, *S. aureus*, *E. faecalis*, and *E. faecium*, and the Spearman *r* values for each comparison.

**Table 1 pathogens-11-00685-t001:** The distribution of *C. striatum*-positive samples by referring diagnosis in 2020–2021. The distribution of 170 *C. striatum*-positive samples by referring diagnoses, as well as the distribution of positive samples based on whether the diagnosis is COVID-19-associated. Diagnoses associated with COVID-19 are in bold and italic. The totals are in bold.

Submitting Diagnoses	Occurrence	Frequency
** *Viral Pneumonia* **	** *14* **	** *8.24%* **
** *Viral Infection* **	** *10* **	** *5.88%* **
Bacterial infections	8	4.71%
** *Dyspnoea* **	** *8* **	** *4.71%* **
Fractura colli femoris medialis	7	4.12%
** *Coronavirus Infection* **	** *7* **	** *4.12%* **
Subarachnoid haemorrhage from the middle cerebral artery	7	4.12%
Septicaemia	6	3.53%
Malignant neoplasm of larynx	6	3.53%
Local infections of the skin and subcutaneous tissues	5	2.94%
Pain	5	2.94%
** *Severe respiratory Failure* **	** *5* **	** *2.94%* **
Chronic renal failure	5	2.94%
Malignant neoplasm of the glottis	4	2.35%
Gastrointestinal bleeding	4	2.35%
Malignant neoplasm of the hypopharynx	4	2.35%
** *Other Viral Pneumonia* **	** *4* **	** *2.35%* **
Aortic (valve) stenosis	3	1.76%
** *COVID-19 with Detected Virus* **	** *3* **	** *1.76%* **
Hodgkin’s disease. lymphocyte predominance	2	1.18%
Congestive heart failure	2	1.18%
Hepatorenal syndrome	2	1.18%
Optic nerve sheath inflammation [neuromyelitis optica devic]	2	1.18%
Contusio cerebri	2	1.18%
Atherosclerosis of the limbal arteries	2	1.18%
Stage III malignancy of the hypopharynx	2	1.18%
Ulceration of the lower limb.	2	1.18%
Haematoma parietis abdominis	1	0.59%
Stage II malignant neoplasm of larynx	1	0.59%
Malignant neoplasm of the upper limb. connective tissue. and soft tissues of the shoulder	1	0.59%
Postoperative subglottic stenosis	1	0.59%
Malignant neoplasm of the root of the tongue	1	0.59%
Tumor of the larynx of uncertain and unknown behavior	1	0.59%
Acute inflammation of the vagina	1	0.59%
Stage III malignant neoplasm of the glottis	1	0.59%
Staphylococcal infection	1	0.59%
Malignant neoplasm of intra-abdominal lymph nodes	1	0.59%
*Staphylococcus aureus* as a cause of other diseases	1	0.59%
Background Retinopathy and retinal lesions	1	0.59%
Oedema	1	0.59%
Severe pansinusitis	1	0.59%
Fractura costarum	1	0.59%
Parotid effusion	1	0.59%
Laryngeal stenosis	1	0.59%
Burn involving less than 10% of the body surface	1	0.59%
Urinary tract infection at unspecified site	1	0.59%
Ear discharge	1	0.59%
Aryepiglottic folds anterior to hypopharynx. malignant swelling III	1	0.59%
Stage III malignant neoplasm of pharynx	1	0.59%
Dermatopolymyositis	1	0.59%
Bacterial pneumonia	1	0.59%
Pain localised to other parts of the abdomen	1	0.59%
Hypertensive heart disease (congestive) without heart failure	1	0.59%
Insulin-dependent diabetes with ocular complications	1	0.59%
Non-Hodgkin lymphoma. large cell (diffuse)	1	0.59%
Cardiac arrest with successful resuscitation	1	0.59%
Malignant tumor of the throat	1	0.59%
Non-defined dementia	1	0.59%
Obesity	1	0.59%
Malignant tumor of the lung	1	0.59%
Postoperative abnormality of the eye and its appendages	1	0.59%
Heart failure	1	0.59%
Non-toxic thyroid nodule	1	0.59%
Other and abdominal pain	1	0.59%
Stage III malignant neoplasm of the posterior wall of the hypopharynx	1	0.59%
Malignancy of the pyriform sinus	1	0.59%
**Total**	**170**	**100.00%**
** *COVID-19-Associated* **	** *51* **	** *30%* **
Not COVID-19-associated	119	70%
**Total**	**170**	**100%**

## Data Availability

Not applicable.

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
