# Peer review of "Corynebacterium striatum—Got Worse by a Pandemic?"

_pathogens, 2022, doi:10.3390/pathogens11060685_

Round 1
Reviewer 1 Report
The quality of the new manuscript has significantly improved compared with the previous version.
The authors have replied to all of my comments and concerns, and included them along the text.
For that, I consider the current version acceptable for publication.
Author Response
Many thanks to Reviewer #1 for once again taking the time to review our work and contributing to the publication.
Reviewer 2 Report
In the manuscript, the authors report the role of Corynebacterium striatum in the nosocomial field. Surprisingly, the prevalence of C. striatum MDR increased from 2012 to 2021, with a notable increase during the COVID-19 pandemic. Furthermore, the multi-drug resistance panel highlights the need for new interventional strategies to counteract nosocomial infections associated with C. striatum strains. The manuscript appears to be interesting and suitable for publication after some minor revisions.
Introduction
-Line 24: consider replacing the phrase "The Corynebacterium species are strictly aerobic, non-spore-forming, club-shaped Gram-positive rods" with "Corynebacteria are non-spore-forming bacteria with considerable pleomorphism, from club-shaped to long slender bacilli"
-Line 40: consider replacing the phrase "Currently, vancomycin, rifampicin, and linezolid are the most active in vitro drugs against the members of this genus" with "Currently, glycopeptides, linezolid, quinupristin/dalfopristin, daptomycin, and tigecycline are the effective drugs available against C. striatum clinical isolates.
Authors should consider that C. striatum multidrug-resistant strains are strong biofilm producers. The latter is an important factor in the development and spread of resistance genes. This aspect should be added in the introduction.
Results
Authors should improve the resolution of the figures.
-Paragraph 2.4: This section is of great importance to support the hypothesis of C. striatum superinfection in patients with COVID-19. The authors should explain these findings in more detail. Also, images 4 A and B are not clear. I would suggest replacing these images with a table or changing the representation mode to make the data more understandable.
Material and Methods
-Line 460. Authors should specify, for each year, how many samples were collected and analyzed
-Line 469. Describe in detail how the antibiogram was performed.
Discussion
-The discussion paragraph should be summarized to understand the key concepts of the manuscript.
-In a recent study Folliero et al. (2022) proposed anthelmintic drugs to counter C. striatum MDR infections. Considering that currently few drugs are available against this pathogen, the authors should mention this manuscript to emphasize the need for new interventional strategies through the drug repurposing ( https://doi.org/10.3390/antibiotics11050651)
Author Response
Thank you again for giving us a chance of revising our manuscript entitled „Corynebacterium striatum - Got Worse by a Pandemic?”.
We appreciate the Reviewers' ideas, which we have attempted to incorporate into our text through proofreading. These changes, we believe, will improve the quality of our work.
Here are the answers point-to-point to the Reviewer’s opinion:
Reviewer #2:
- -Line 24: consider replacing the phrase "The Corynebacterium species are strictly aerobic, non-spore-forming, club-shaped Gram-positive rods" with "Corynebacteria are non-spore-forming bacteria with considerable pleomorphism, from club-shaped to long slender bacilli"
Answer: Thank you very much for your suggestion. The text has been amended accordingly.
- -Line 40: consider replacing the phrase "Currently, vancomycin, rifampicin, and linezolid are the most active in vitro drugs against the members of this genus" with "Currently, glycopeptides, linezolid, quinupristin/dalfopristin, daptomycin, and tigecycline are the effective drugs available against C. striatum clinical isolates.”
Answer: Once again, thank you so much for your proposal. The wording has been updated to reflect this.
- Authors should consider that C. striatum multidrug-resistant strains are strong biofilm producers. The latter is an important factor in the development and spread of resistance genes. This aspect should be added in the introduction.
Answer: Thank you very much for drawing our attention to this aspect. We have included this in the Introduction.
- Authors should improve the resolution of the figures.
Answer: As defined in the Instructions for Authors (Pathogens | Instructions for Authors (mdpi.com)), the resolution of the figures must be 300 dpi or higher. We used 600 dpi for all figures. If you have any further suggestions on which figures to increase the resolution and by how much, we would appreciate it.
- -Paragraph 2.4: This section is of great importance to support the hypothesis of C. striatum superinfection in patients with COVID-19. The authors should explain these findings in more detail. Also, images 4 A, and B are not clear. I would suggest replacing these images with a table or changing the representation mode to make the data more understandable.
Answer: Thank you for bringing this to our attention. Accordingly, we have included a section highlighting this in the Discussion, and Fig. 4 A and B have been replaced by Table 1 A and B.
- -Line 460. Authors should specify, for each year, how many samples were collected and analyzed.
Answer: Thank you for bringing this opportunity to our attention. However, we should note that this information can be found in Fig. 1. For this reason, we do not consider it appropriate to repeat this in the Materials and methods section.
- -Line 469. Describe in detail how the antibiogram was performed.
Answer: We are grateful to you for drawing our attention to this shortcoming. We have modified the relevant section of the Materials and methods accordingly.
- -The discussion paragraph should be summarized to understand the key concepts of the manuscript.
Answer: Unfortunately, despite our best efforts, we were not able to write the Discussion in a shorter version, since it was necessary to include quite a lot of results. However, as you suggested, the emphasis on key concepts is perhaps better in this edition. That said, if you have any suggestions for revising the Discussion, they are welcome.
- -In a recent study Folliero et al. (2022) proposed anthelmintic drugs to counter C. striatum MDR infections. Considering that currently few drugs are available against this pathogen, the authors should mention this manuscript to emphasize the need for new interventional strategies through the drug repurposing (https://doi.org/10.3390/antibiotics11050651)
Answer: Thank you very much for bringing this important publication to our attention. As suggested, we have included it in the Discussion and References.
Thank you very much for the reviewing work.
Reviewer 3 Report
Orosz and co-authors addressed an interesting issue revealing the possible correlation between COVID-19 pandemic, with subsequently increasing hospitalization, and the arose of Corynebacterium striatum isolation in patients.
Unfortunately, the overall scientific quality of the paper is undermined by several issues, the major regarding the resistance of C. striatum to linezolid, something that even the authors said it have been never reported before in literature. In their paper (Fig. 5A), authors reported the isolation of linezolid resistance strains only between 2017 and 2020, it would be interesting to understand how many strains were actually linezolid resistant in 2017, in 2018... and what they thought about the disappearing of this resistance in 2021. Nonetheless, in their prediction (Fig. 7) they conjectured that even in 2021/2022 linezolid resistance strains should be isolated.
All these results are affected also by the lack of explanations about the methods employed for the identification of the bacteria and their AST.
Minor comments:
- what did the authors mean with superficial and deep samples? From which tissues and organs?
- in the introduction they should state that this is a retrospective study.
- in the figures authors should be more clear writing "number of C. striatum positive samples" instead of "number of positive samples". Also, pre-COVID and COVID should be written as COVID-19 (but it is something less important).
- Figure 3, did COVID care ward exist before 2020? It seems unlikely.
- Figure 4, why the two totals are different in the graph? In the text authors talk about 170 samples, what about the missing 55 samples in the A graph? Were they representative of single cases?
- Can the authors explain how the rifampicin resistance could be transferred by S. aureus to C. striatum? It should be given by a mutation in the rpoB gene.
- Why do the authors use different statistical analysis (welch's test OR mann-Whitney, Chi-square OR fisher's) without stating when they choose a test or the other one?
- Finally, there are some little mistakes as Eneterococcus (line 240)
Author Response
Thank you again for giving us a chance of revising our manuscript entitled „Corynebacterium striatum - Got Worse by a Pandemic?”.
We appreciate the Reviewers' ideas, which we have attempted to incorporate into our text through proofreading. These changes, we believe, will improve the quality of our work.
Here are the answers point-to-point to the Reviewer’s opinion:
Reviewer #3:
- Unfortunately, the overall scientific quality of the paper is undermined by several issues, the major regarding the resistance of C. striatum to linezolid, something that even the authors said it have been never reported before in literature. In their paper (Fig. 5A), authors reported the isolation of linezolid resistance strains only between 2017 and 2020, it would be interesting to understand how many strains were actually linezolid resistant in 2017, in 2018... and what they thought about the disappearing of this resistance in 2021.
Answer: We understand the Reviewer's concerns. It has been indicated that this is a phenomenon not previously reported. The linezolid resistance was observed in 39 isolates from 28 patients between 2017-2020 (see the graph below for details).
Prevalence of linezolid-resistant C. striatum strains at the Clinical Center of the University of Szeged, 2017-2020.
The authors believe that this is beyond the limit of the probability of measurement error. Unfortunately, however, since the strains were not stored at that time, and since no one looked into the literary background of this resistance type, we were unable to investigate the exact background of this trend afterward.
Another legitimate question is why the strains disappeared in 2021? If we compare the unit distribution of the linezolid-resistant strains with that of all C. striatum isolates shown, we can see that there are significant differences (see the table below and Fig. 3 in the manuscript). This linezolid-resistant strain was most commonly observed among Emergency room patients, all of whom are outpatients, and only in 2020 did we see a small accumulation among patients in the 2nd Dept. of Internal Medicine. This suggests that this strain may have been introduced from the outside rather than spreading widely within the units of the clinical center. Since outpatient traffic in Hungary dropped significantly in 2020 due to the lockdown caused by the COVID-19 pandemic, it is possible that this strain could not spread. Although the presence of this kind of isolate was also detected in the Otorhinolaryngology unit, it seems to have been replaced by another strain in 2019/2020, which is thought to have spread in the COVID-19 care units.
|
Distribution of linezolid-resistant C. striatum strains |
|||||
|
Units |
2017 |
2018 |
2019 |
2020 |
Total |
|
Emergency room |
3 |
2 |
0 |
4 |
9 |
|
2nd Dept. Of Internal Medicine |
1 |
1 |
0 |
5 |
7 |
|
Otorhinolaryngology |
0 |
3 |
3 |
1 |
7 |
|
Surgery |
0 |
3 |
1 |
0 |
4 |
|
3rd ICU |
2 |
1 |
0 |
0 |
3 |
|
Ophthalmology |
1 |
0 |
0 |
0 |
1 |
|
Chronic care |
1 |
0 |
0 |
0 |
1 |
|
Traumatology |
1 |
0 |
0 |
0 |
1 |
|
2nd ICU |
1 |
0 |
0 |
0 |
1 |
|
Stroke |
0 |
1 |
0 |
0 |
1 |
|
Rheumatology |
0 |
1 |
0 |
0 |
1 |
|
Pediatrics |
0 |
0 |
1 |
0 |
1 |
|
Dermatology |
0 |
0 |
1 |
0 |
1 |
|
Oncology |
0 |
0 |
1 |
0 |
1 |
We speculate that there may have been a difference in respiratory pathogenicity between the two strains. This is supported by the observation that linezolid-resistant isolates were mainly derived from blood culture (see the table below).
|
Sample types |
2017 |
2018 |
2019 |
2020 |
Total |
|
blood culture |
4 |
1 |
0 |
9 |
14 |
|
trachea |
2 |
4 |
4 |
1 |
11 |
|
wound |
2 |
4 |
0 |
0 |
6 |
|
abscess |
0 |
1 |
1 |
0 |
2 |
|
urine |
0 |
1 |
1 |
0 |
2 |
|
eye |
1 |
0 |
0 |
0 |
1 |
|
surgical |
1 |
0 |
0 |
0 |
1 |
|
middle ear |
0 |
1 |
0 |
0 |
1 |
|
throat |
0 |
0 |
1 |
0 |
1 |
We suspect that as the COVID-19 pandemic progressed and the number of patients ventilated increased, this strain was displaced by another isolate that is sensitive to linezolid and demonstrates primarily respiratory pathogenicity (eg. by more ability to form biofilms). During 2020 and 2021 this strain could disseminate throughout our Clinical Center, but mainly in the COVID-19 wards.
Of course, the authors are aware that this is pure speculation, which is why we have not devoted much space to it in the manuscript. As we are now paying more attention to C. striatum, if a linezolid-resistant isolate reappears, it will be subjected to genetic testing.
The authors hope that by sharing all this data we have been able to allay the Reviewer's doubts but ask for understanding on the issue as we do not have any more information on the linezolid-resistant C. striatum strain.
- All these results are affected also by the lack of explanations about the methods employed for the identification of the bacteria and their AST.
Answer: Thank you to the Reviewer for drawing our attention to this shortcoming. Accordingly, we have added a reference to the methodology for resistance testing in the Materials and methods section. (See also paragraph #7 of the response to Reviewer #2.)
- - what did the authors mean with superficial and deep samples? From which tissues and organs?
Answer: We grouped superficial samples from the skin surface and superficial wounds, where C. striatum may occur as a member of the normal flora. Isolates from abscesses, drains, and surgical specimens, where the chances of contamination were lower, were called deep samples. The superficial/deep distinction is therefore mainly a differentiation according to the chance of contamination.
- - in the introduction they should state that this is a retrospective study.
Answer: We agree with the Reviewer. We have modified the Introduction section on line 65 accordingly.
- - in the figures authors should be more clear writing "number of C. striatum positive samples" instead of "number of positive samples". Also, pre-COVID and COVID should be written as COVID-19 (but it is something less important).
Answer: Thank you for this suggestion. We have modified the following figures accordingly: Figs. 1, 2, 4 (5 in the previous manuscript version), 7 (8 in the previous manuscript version).
- - Figure 3, did COVID care ward exist before 2020? It seems unlikely.
Answer: Thank you to the Reviewer for pointing out this error. Accordingly, we have modified the affected cells on the figure in question by leaving them empty in the years before 2020.
- - Figure 4, why the two totals are different in the graph? In the text authors talk about 170 samples, what about the missing 55 samples in the A graph? Were they representative of single cases?
Answer: At the suggestion of Reviewer #2, the figure in question was replaced by a table with data from 170-170 samples in each part (see paragraph #5 in answer to Reviewer #2).
- - Can the authors explain how the rifampicin resistance could be transferred by S. aureus to C. striatum? It should be given by a mutation in the rpoB gene.
Answer: We fully agree with Referee, and our data show that there is no correlation between the co-occurrence of the two species and the increase in rifampicin resistance. However, given the frequent use of rifampicin-containing drugs in the treatment of e.g. cannula infections caused by S. aureus at our Clinical Center, this association may have contributed to the spread of resistance by increasing selection pressure. However, a discussion of this is beyond the scope of the present work.
- - Why do the authors use different statistical analysis (welch's test OR mann-Whitney, Chi-square OR fisher's) without stating when they choose a test or the other one?
Answer: As we have stated in the Statistical analysis subsection of the Materials and methods: „Welch’s t-test or Mann-Whitney U test was used to compare the means of populations. Chi-square test or Fischer’s exact test was used for frequency distributions.” Different statistical methods were used accordingly for each analysis.
- - Finally, there are some little mistakes as Eneterococcus (line 240)
Answer: We have modified the text accordingly.
Thank you very much for taking the time to review our work.